# Analysis of close associations of uropod-associated proteins in human T-cells using the proximity ligation assay

Tommy Baumann, Sarah Affentranger and Verena Niggli

Institute of Pathology, University of Bern, Bern, Switzerland

## ABSTRACT

We have shown previously that the raft-associated proteins flotillin-1 and -2 are rapidly recruited to the uropods of chemoattractant-stimulated human neutrophils and T-cells and are involved in cell polarization. Other proteins such as the adhesion receptor PSGL-1, the actin-membrane linker proteins ezrin/radixin/moesin (ERM) and the signaling enzyme phosphatidylinositol-4-phosphate 5-kinase type Iγ90 (PIPKIγ90) also accumulate in the T-cell uropod. Using the in situ proximity ligation assay (PLA) we now have investigated putative close associations of these proteins in human freshly isolated T-cells before and after chemokine addition. The PLA allows in situ subcellular localization of close proximity of endogenous proteins at single-molecule resolution in fixed cells. It allows detection also of weaker and transient complexes that would not be revealed with co-immunoprecipitation approaches. We previously provided evidence for heterodimer formation of tagged flotillin-1 and -2 in T-cells before and after chemokine addition using fluorescence resonance energy transfer (FRET). We now confirm these findings using PLA for the endogenous flotillins in fixed human T-cells. Moreover, in agreement with the literature, our PLA findings confirm a close association of endogenous PSGL-1 and ERM proteins both in resting and chemokine-activated human T-cells. In addition, we provide novel evidence using the PLA for close associations of endogenous activated ERM proteins with PIPKIγ90 and of endogenous flotillins with PSGL-1 in human T-cells, before and after chemokine addition. Our findings suggest that preformed clusters of these proteins coalesce in the uropod upon cell stimulation.

## INTRODUCTION

Chemotactic migration of the highly motile T-cells is indispensable for the fulfillment of their physiological functions. T-cell polarization is a prerequisite for directional migration. Polarization of leukocytes requires segregation and activation of specific signaling and cytoskeletal molecules in the retracting rear (uropod) and motile forward moving part (front) of the cells. Localized positive feedback loops and inhibitory effects of front signaling pathways on rear signaling and vice versa are thought to reinforce this structural and biochemical polarization (*Friedl & Weigelin, 2008*; *Bagorda & Parent, 2008*). Plasma membrane microdomains ("rafts") have been implicated to stabilize polarity of migrating leukocytes (*Manes & Viola, 2006*).

Corresponding author
Verena Niggli,
verena.niggli@pathology.unibe.ch

The uropod is a plasma membrane protrusion located in the rear of migrating leukocytes that contains specific organelles along with cytoskeletal, adhesion and signaling proteins such as activated phosphorylated ezrin/radixin/moesin proteins (P-ERM), the adhesion receptor P-selectin glycoprotein-1 (PSGL-1) and phosphatidylinositol-4-phosphate 5-kinase type I (PIPKI) γ90 (*Lokuta et al., 2007*; *Sánchez-Madrid & Serrador, 2009*; *Mathis et al., 2013*). Flotillins, membrane microdomain scaffolding proteins, are also enriched in leukocyte uropods and are involved in uropod formation (*Ludwig et al., 2010*; *Rossy et al., 2009*; *Affentranger et al., 2011*; *Baumann, Affentranger & Niggli, 2012*). The uropod may be especially required for T-cell migration through constricted spaces (*Soriano et al., 2011*).

We currently study the mechanisms of targeting of specific proteins to the T-cell uropod.

Transfection of human freshly isolated T-cells with a dominant-negative mutant of flotillin-2 impaired cell polarization and uropod capping of endogenous flotillin-1, PSGL-1 and GFP-tagged PIPKIγ90, indicating a functional role of flotillins in structuring the uropod (*Affentranger et al., 2011*; *Mathis et al., 2013*). Moreover, expression of constitutively active ezrin in freshly isolated T-cells induced capping of flotillins and PSGL-1, and that of a dominant-negative ezrin mutant impaired flotillin and PSGL-1 capping, suggesting a scaffolding role also for P-ERM (*Martinelli et al., 2013*). Similarly overexpression of PIPKIγ87, a naturally occurring isoform lacking the last 26 amino acids, which does not locate to the uropod, suppresses T-cell uropod formation and impairs capping of uropod proteins such as flotillins (*Mathis et al., 2013*).

We have now studied in situ protein-protein interactions in human T-cells fixed before and after chemokine addition, using the proximity ligation assay (PLA), in order to obtain insight into the molecular processes involved in T-cell uropod formation. The PLA allows in situ subcellular localization of close proximity of proteins at single-molecule resolution (*Söderberg et al., 2006*). It also allows detection of weaker and transient complexes that would not be revealed with co-immunoprecipitation approaches. In contrast to fluorescence resonance energy transfer (FRET), which involves expression of tagged proteins, PLA allows analysis of complexes of unmodified endogenous proteins. We analyzed selected interactions of T-cell uropod-located proteins for which high quality antibodies working well in immunofluorescence are available. We focused on flotillins, PSGL-1, activated ERM proteins and PIPKIγ90. We provide novel data indicating close proximity of P-ERM and PIPKIγ90 and of flotillins and PSGL-1 in T-cells before and after chemokine addition. As expected from previous data (*Ivetic & Ridley, 2004*), PSGL-1 also closely associates with P-ERM. Associations of flotillins with P-ERM or with PIPKIγ90 appear to be less extensive.

## MATERIALS AND METHODS

### Materials and suppliers

Stromal cell derived factor 1 (SDF-1): Peprotech, Paris, France.

Bovine serum albumin (BSA): Serva, Germany.

Gey's solution contained 138 mM NaCl, 6 mM KCl, 100 μM EGTA, 1 mM $Na_2HPO_4$, 5 mM $NaHCO_3$, 5.5 mM glucose and 20 mM Hepes (pH 7.4).

## Antibodies

Monoclonal murine antibodies directed against flotillin-2 (Cat. No. E35820) and PSGL-1 (Cat. No. 556053) were obtained from Transduction Laboratories/BD Pharmingen, Germany. A polyclonal rabbit antibody directed against phospho ezrin (Thr567)/radixin (Thr564)/moesin (Thr558) (Cat. No. 3141) was from Cell Signaling Technology. Polyclonal rabbit antibodies directed against flotillin-2 (Cat. No. F1680) or flotillin-1 (Cat. No. F1180), a monoclonal murine antibody recognizing the hemaglutinin tag (HA) (clone HA-7) and a FITC-conjugated antibody directed against murine IgG (Cat. No. F5387) were obtained from Sigma. A monoclonal murine antibody specifically recognizing $\beta$-cytoplasmic actin was kindly provided by C Chaponnier (*Dugina et al., 2009*).

## Construct

For preparation of N-terminally HA-tagged PIPKIγ90, a construct coding for wild-type PIP5KIγ661 N-terminally tagged with EGFP in a pcDNA3.1 vector (*Lokuta et al., 2007*) was used as a PCR template. Then the PCR product was cloned into the phCMV2 Vector (Genlantis).

## Isolation of human T-lymphocytes

Resting T-lymphocytes were isolated from buffy coats of healthy donor blood using the Pan T Cell Isolation Kit II (Miltenyi Biotec) and separation on LS columns (Miltenyi Biotec) as described previously (*Martinelli et al., 2013*). The buffy coats were obtained from the Central Laboratory of the Swiss Red Cross, Bern, Switzerland. The resulting cell suspension contained >95% T-lymphocytes as assessed using anti-CD3 staining. The cells were used after overnight incubation in RPMI with 10% FCS at 37°C and 5% $CO_2$.

## Immunofluorescence staining

T-cells were incubated as described in the figure or table legends, followed by fixation with TCA or PFA and staining for the indicated proteins as described (*Affentranger et al., 2011*).

## Transient transfections of T-lymphocytes

For transfections, 3–6 × $10^6$ freshly isolated T-lymphocytes were resuspended in 100 μl human T cell nucleofector solution (Amaxa, Köln, Germany) diluted 1:2 with PBS and 1 μg of plasmid DNA was added. Then, the cell suspension was transferred to a cuvette and nucleofection was carried out (Amaxa Nucleofector, program U-14). Immediately, 500 μl of medium with 20% FCS was added and the cells were transferred to a prewarmed 12-well plate containing 2.5 ml of medium with 20% FCS, followed by incubation at 37°C in a $CO_2$ incubator for 4 h. Transfected cells were subsequently washed, resuspended in Gey's solution and used for experiments.

## Proximity ligation assay

The in situ PLA (kit obtained from Olink Bioscience; www.olink.com) was used to detect protein-protein interactions in fixed cells. Nontransfected or transfected cells were treated without or with SDF-1, as described in the figure legends, followed by

fixation with TCA or PFA as described (*Affentranger et al., 2011*), incubation with primary antibodies, incubation with the PLA probes (anti murine and anti-rabbit IgG antibodies conjugated with oligonucleotides), ligation and amplification according to the manufacturer's instructions. Imaging was performed on fixed samples with a confocal laser scanning microscope Olympus Fluoview FV1000-IX81, 60× oil immersion objective. For determination of the fraction of cells with at least one red fluorescent dot per cell 100 cells were evaluated per sample and experiment by microscopical analysis. For the evaluation of the specific number of dots per cell and the number of dots per uropod, pictures of representative optical sections of cells with positive PLA were used (the numbers of the cells analyzed per experiment and condition are indicated in the text).

## RESULTS

### Interaction of flotillin-1 and -2 in freshly isolated human T-cells analyzed with the proximity ligation assay

Endogenous flotillin-1 and -2 showed marked colocalization in resting and chemokine-stimulated freshly isolated human T-cells (Fig. 1), as shown previously (*Affentranger et al., 2011*; *Baumann, Affentranger & Niggli, 2012*). Resting cells were mainly spherical, with a punctate, membrane-associated location of both flotillin-1 and -2. Upon stimulation of cells with the chemokine SDF-1 for 15 min, the majority of the cells polarized, correlating with exclusive location of both flotillins at the plasma membrane at the tips of the uropods (Fig. 1A). Our previous results using FRET in cells transfected with tagged flotillin-1 and -2 suggest that preformed flotillin-1 and -2 heterooligomers coalesce in chemokine-stimulated T-cells (*Baumann, Affentranger & Niggli, 2012*). In the present study, we have investigated interactions of endogenous flotillin-1 and -2, using the PLA, as a positive control. The in situ PLA is based on fixed cells incubated with primary murine and rabbit antibodies reacting for example with flotillin-1 and -2 respectively, followed by incubation of samples with modified secondary antibodies reacting with murine or rabbit IgG antibodies. These secondary antibodies are conjugated with oligonucleotides (PLA probe minus and PLA probe plus). Annealing of the probes occurs when the target proteins are in close proximity (less than 30–40 nm distance), which then initiates the amplification. The amplicons can be detected as red dots by fluorescence microscopy (*Söderberg et al., 2006*). We obtained a strong PLA signal in the majority of the T-cells using primary murine and rabbit antibodies to flotillin-2 and -1 respectively. This signal was located randomly along the plasma membrane in resting cells (range: 1–11 dots per cell; mean: $4 \pm 1$ dots per cell; 58 cells analyzed in 3 experiments), and aligned linearly around the entire border of the uropod (at least 4 dots per uropod) in 65% ($n = 2$; 86 cells analyzed) of the stimulated cells, corresponding to the location of endogenous flotillins (Fig. 1B; top panels: lower magnification; lower panels; higher magnification). These data are in agreement with our FRET studies indicating heterooligomerization of tagged flotillin-1 and -2 (*Baumann, Affentranger & Niggli, 2012*). Very few cells with one red dot corresponding to a positive PLA reaction per cell were detected when the samples were only incubated with the flotillin-1 antibody (Fig. 1C).

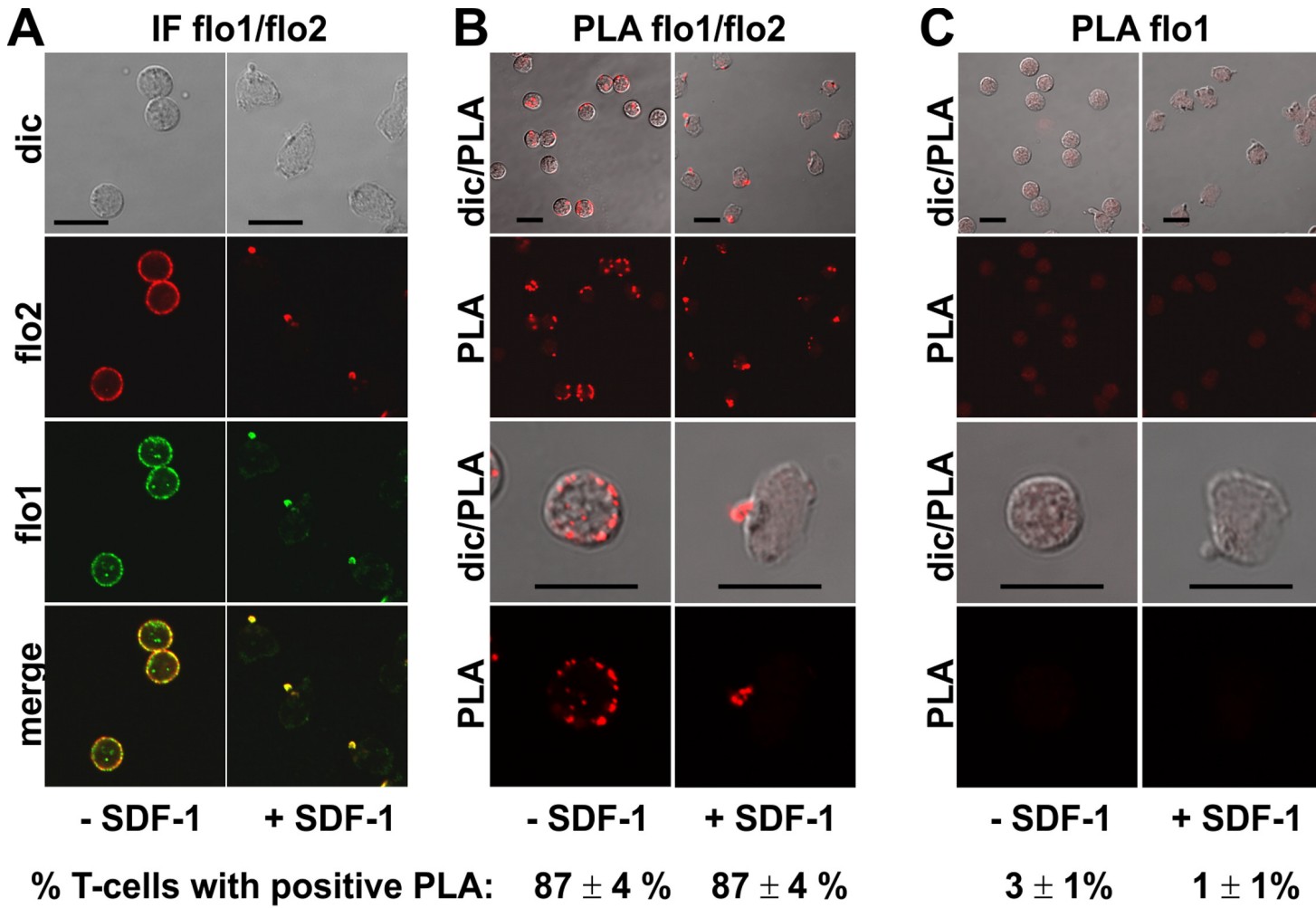

| | | | |
|---|---|---|---|
| **% T-cells with positive PLA:** | 87 ± 4 % | 87 ± 4 % | 3 ± 1% |
| | | | 1 ± 1% |

**Figure 1** **Interaction of flotillin-1 and -2 in human T-cells studied with PLA.** (A, B) T-cells were preincubated for 30 min at 37°C, followed by a further incubation for 15 min without or with 40 ng/ml SDF-1, fixation with TCA and staining for endogenous flotillin-1 (flo1) (rabbit polyclonal antibody) and flotillin-2 (flo2) (monoclonal murine antibody), followed by (A) fluorescently labeled anti-murine and anti-rabbit IgG second antibodies (IF) or (B) PLA probes minus and plus, ligation and amplification. (C) For negative controls, T-cells were treated as described for (B) except that the anti-flotillin-2 antibody was omitted. For (B) and (C), the top panels are overviews at lower magnification whereas in the lower panels single cells are shown at higher magnification. The pictures are representative of 3 experiments. The percentage of cells with one or more red fluorescent dots per cell was determined for 100 cells per sample and experiment (mean ± sem of 3 experiments). Note that the majority of the cells incubated with both flo1 and flo2 antibodies exhibited several dots per cell, whereas for controls only incubated with flo1 antibody, maximally 1 dot per cell occurred. Scale bars, 10 µm.

## Interactions of P-ERM with PSGL-1 and of flotillins with PSGL-1 and P-ERM in T-cells studied using PLA

We studied in situ interactions of endogenous flotillins with the adhesion receptors PSGL-1 and activated phosphorylated ERM (P-ERM) proteins, and of PSGL-1 with P-ERM in fixed human T-cells. Immunofluorescence pictures indeed show partial or extensive colocalization of PSGL-1 with P-ERM (Fig. 2A) and of flotillins with PSGL-1 (Fig. 3A) and P-ERM (Fig. 4A) in resting T-cells and in the uropod of stimulated T-cells. We then analysed whether these colocalizations correlate with close interactions using PLA

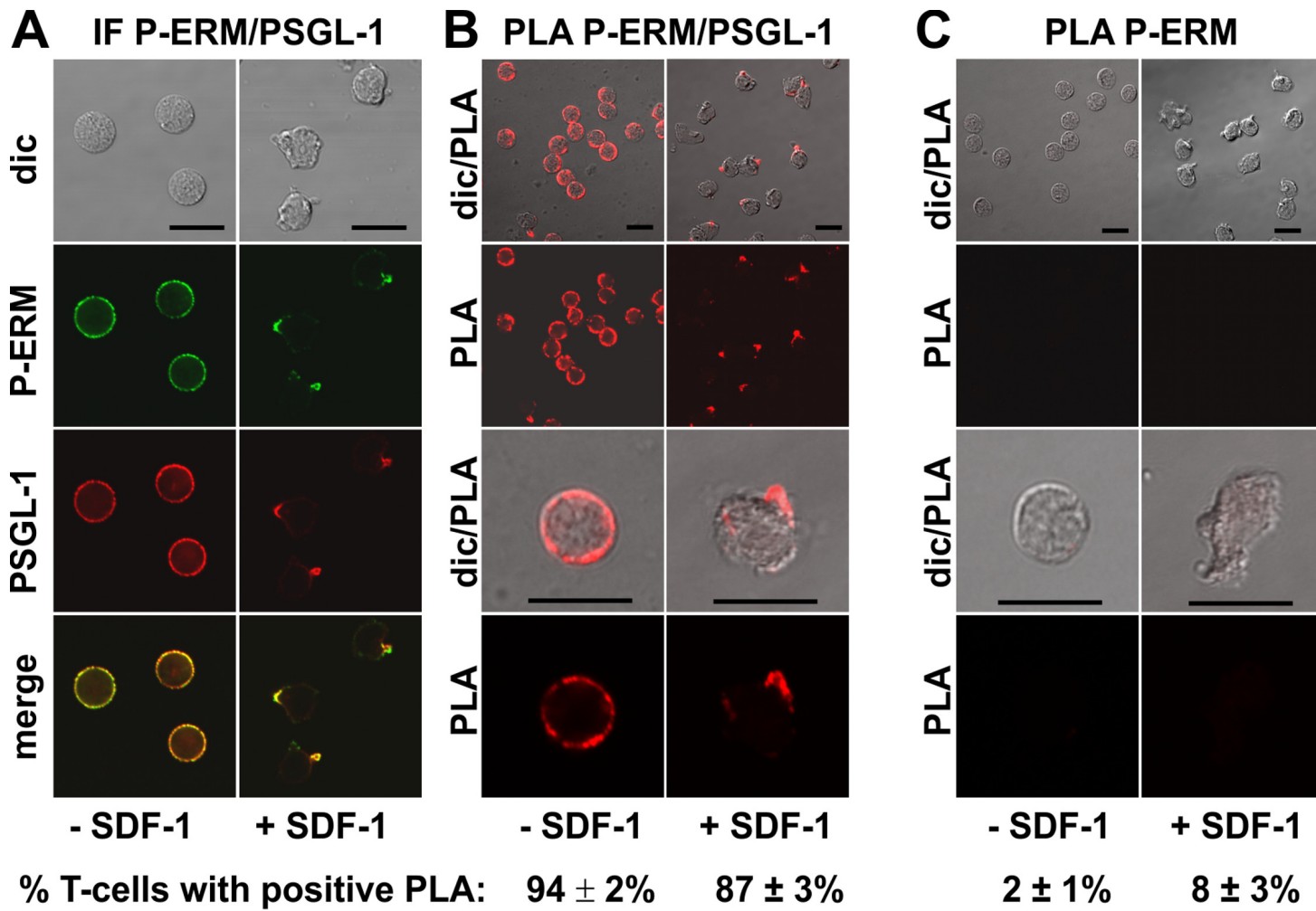

| | - SDF-1 | + SDF-1 | | - SDF-1 | + SDF-1 |
|---|---|---|---|---|---|
| **% T-cells with positive PLA:** | 94 ± 2% | 87 ± 3% | | 2 ± 1% | 8 ± 3% |

**Figure 2** **Interaction of PSGL-1 and P-ERM in human T-cells studied with PLA.** (A, B) T-cells were preincubated for 30 min at 37°C, followed by a further incubation for 15 min without or with 40 ng/ml SDF-1, fixation with TCA and staining for endogenous PSGL-1 (monoclonal murine antibody) and P-ERM (polyclonal rabbit antibody), followed by (A) fluorescently labeled anti-murine and anti-rabbit IgG second antibodies (IF) or (B) PLA probes minus and plus, ligation and amplification. (C) For negative controls, T-cells were treated as described for (B), except that the anti-PSGL-1 antibody was omitted. For (B) and (C), the top panels are overviews at lower magnification whereas in the lower panels single cells are shown at higher magnification. The pictures are representative of 3 experiments. The percentage of cells with one or more red fluorescent dots per cell, indicating positive PLA, was determined for 100 cells per sample and experiment (mean ± sem of 3 experiments). Scale bars, 10 μm.

in human T-cells. As a positive control we studied the well established direct interaction between PSGL-1 and P-ERM using primary antibodies specifically recognizing PSGL-1 and P-ERM respectively which work well in immunofluorescence (Fig. 2A). As expected from previous findings (*Ivetic & Ridley, 2004*), we obtained positive PLA signals for PSGL-1 and P-ERM in 94 ± 2% of resting and 87 ± 3% ($n = 3$) of chemokine-activated cells (Fig. 2B). In resting cells the dots indicating close proximity of the proteins were randomly located at the cell periphery (range: 4–20 dots per cell; mean: 12 ± 1 dot per cell, analysed in 60 cells derived from 3 experiments). In stimulated cells the dots lined the entire border of the uropod in 55 ± 5% ($n = 3$) of the polarized PLA-positive cells (a total of 248 cells analysed). The remainder of the polarized PLA-positive cells featured 1–2 dots/uropod.

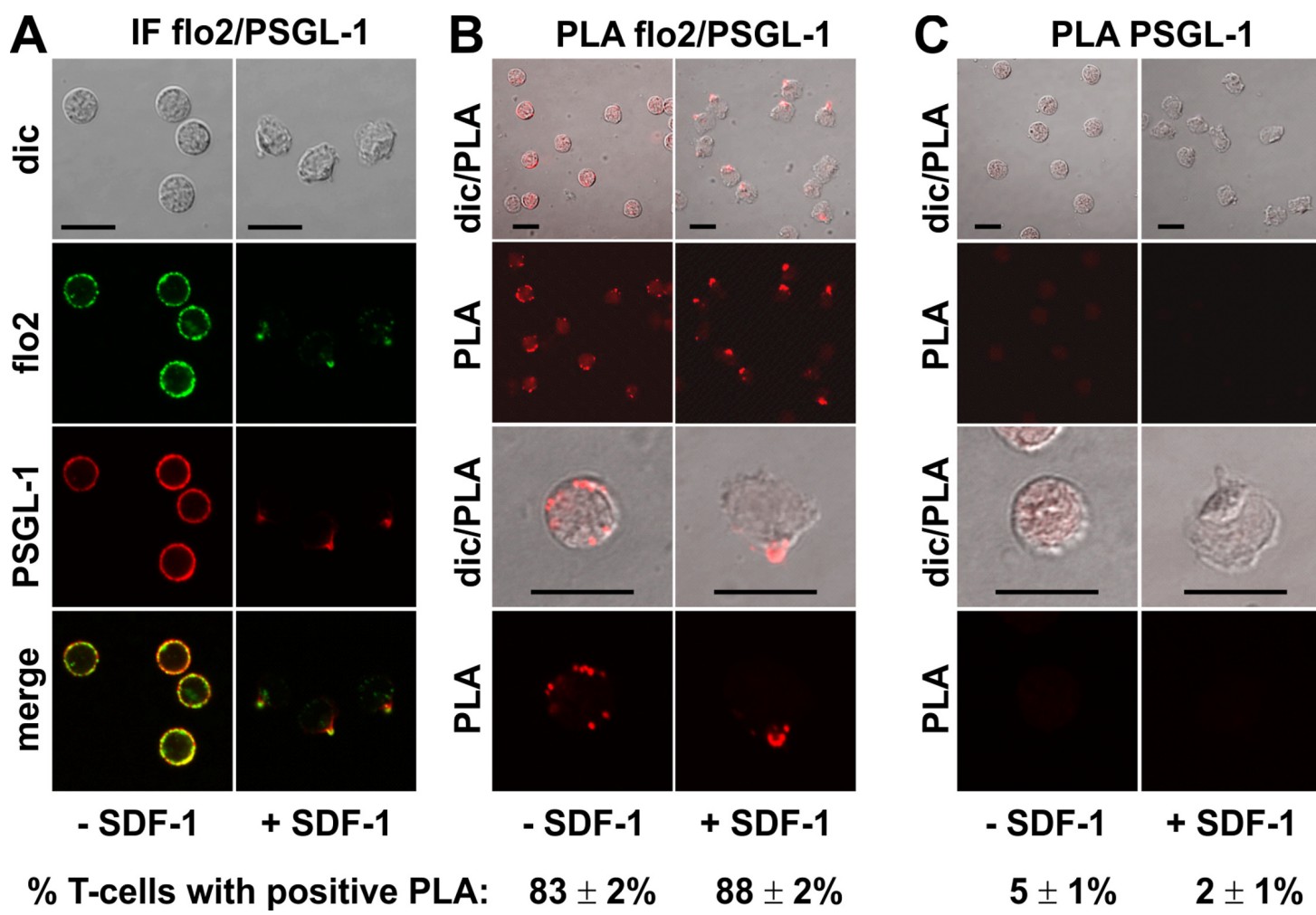

| | | | |
|---|---|---|---|
| **% T-cells with positive PLA:** | $83 \pm 2\%$ | $88 \pm 2\%$ | $5 \pm 1\%$ | $2 \pm 1\%$ |

**Figure 3** **Interaction of PSGL-1 and flotillin-2 in human T-cells studied with PLA.** (A, B) T-cells were preincubated for 30 min at 37°C, followed by a further incubation for 15 min without or with 40 ng/ml SDF-1, fixation with TCA and staining for endogenous PSGL-1 (monoclonal murine antibody) and flotillin-2 (polyclonal rabbit antibody), followed by (A) fluorescently labeled anti-murine and anti-rabbit IgG second antibodies (IF) or (B) PLA probes minus and plus, ligation and amplification. (C) For negative controls, T-cells were treated as described for (B), except that the anti-flotillin-2 antibody was omitted. For (B) and (C), the top panels are overviews at lower magnification whereas in the lower panels single cells are shown at higher magnification. The pictures are representative of 3 experiments. The percentage of cells with one or more red fluorescent dots per cell was determined for 100 cells per sample and experiment (mean $\pm$ sem of 3 experiments). Scale bars, 10 μm.

A negative control where the samples were only incubated with the P-ERM antibody is shown in Fig. 2C.

A positive PLA reaction was also observed for PSGL-1 and flotillin-2 in resting and chemokine-activated T-cells, confirming and extending the data obtained in human neutrophils using co-immunoprecipitation of flotillin-2 and PSGL-1 (*Rossy et al., 2009*). Here we obtained positive PLA signals in $83 \pm 2\%$ ($n = 4$) of the resting cells and $88 \pm 2\%$ ($n = 4$) of the chemokine-stimulated T-cells (Fig. 3B), with fluorescent dots located at the plasma membrane of the resting cells (range: 1–11 dots per cell; mean: $4 \pm 1$ dots per cell analysed in 30 cells derived from 3 experiments), and along the entire uropod border in 67% ($n = 2$; 198 cells analysed) of the polarized, PLA-positive stimulated cells. The

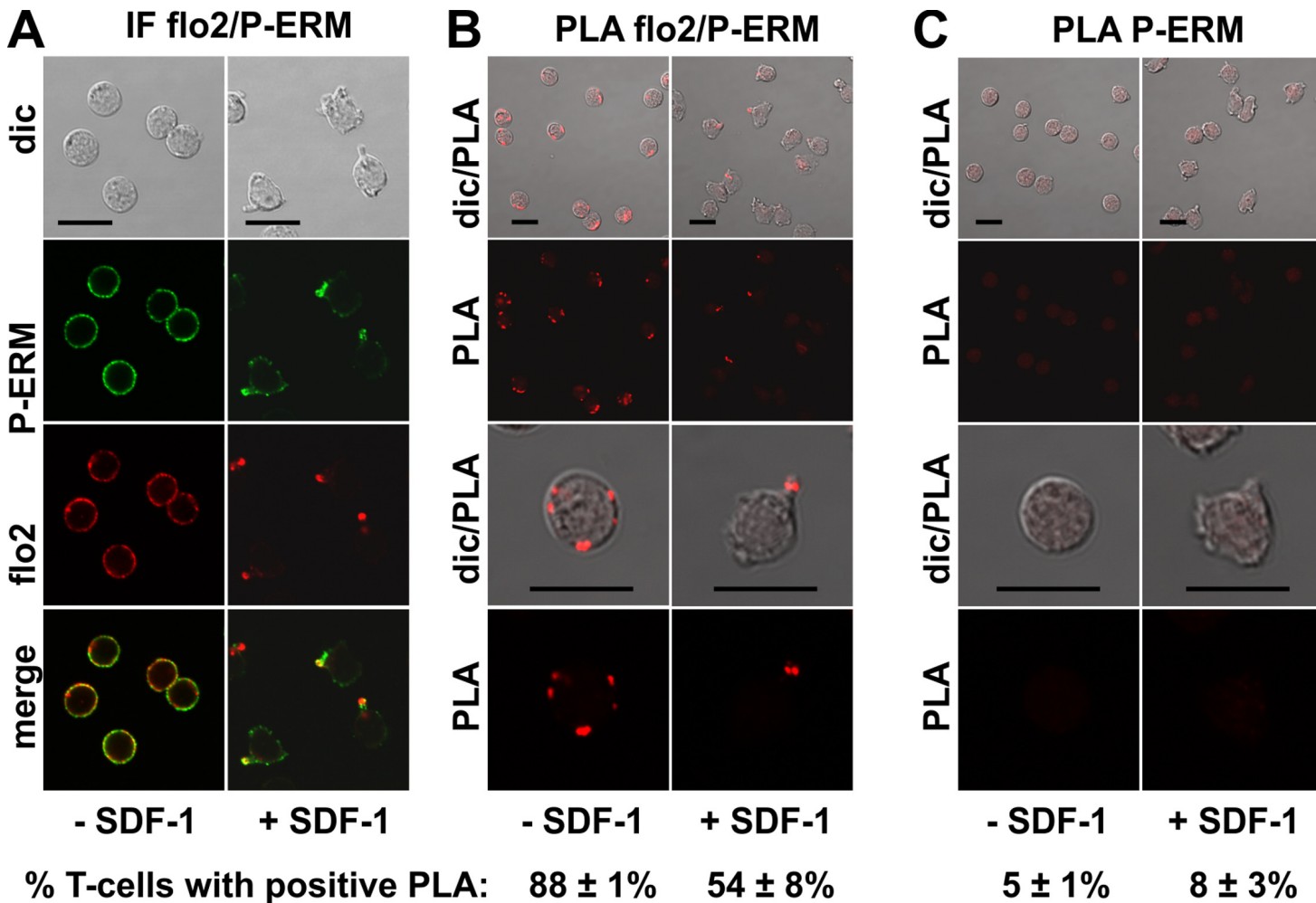

**A** IF flo2/P-ERM
**B** PLA flo2/P-ERM
**C** PLA P-ERM

| | − SDF-1 | + SDF-1 | | − SDF-1 | + SDF-1 |
|---|---|---|---|---|---|
| % T-cells with positive PLA: | 88 ± 1% | 54 ± 8% | | 5 ± 1% | 8 ± 3% |

**Figure 4** **Interaction of P-ERM and flotillin-2 in human T-cells studied with PLA.** (A, B) T-cells were preincubated for 30 min at 37°C, followed by a further incubation for 15 min without or with 40 ng/ml SDF-1, fixation with TCA and staining for endogenous flotillin-2 (monoclonal murine antibody) and P-ERM (polyclonal rabbit antibody), followed by (A) fluorescently labeled anti-murine and anti-rabbit IgG second antibodies (IF) or (B) PLA probes minus and plus, ligation and amplification. (C) For negative controls, T-cells were treated as described for (B), except that the anti-flotillin-2 antibody was omitted. For (B) and (C), the top panels are overviews at lower magnification whereas in the lower panels single cells are shown at higher magnification. The pictures are representative of 3 experiments. The percentage of cells with one or more red fluorescent dots per cell was determined for 100 cells per sample and experiment (mean ± sem of 3 experiments). Scale bars, 10 μm.

remainder of the polarized PLA-positive cells featured 1–2 dots/uropod. Negative controls with only the anti-PSGL-1 antibody are shown in Fig. 3C.

The PLA of flotillin-2 and P-ERM was also positive in 88 ± 1% of the resting T-cells (range: 1–6 dots per cell; mean: 3 ± 1 dots per cell analysed in 59 cells derived from 2 experiments), and in 54 ± 8% of the stimulated cells. Especially in the stimulated cells the number of dots per cell was clearly lower as compared to the result obtained for PSGL-1 and flotillin, indicating weaker, possibly transient interactions (Fig. 4B). Only 15 ± 7% ($n = 3$) of polarized, PLA-positive cells featured dots lining the border of the entire uropod (a total of 261 cells analysed). The remainder of the polarized PLA-positive cells featured 1–2 dots /uropod. Negative controls using only the anti-P-ERM antibody are shown in Fig. 4C.

As a negative control we also applied the PLA assay to $\beta$-actin and flotillin-2 (Fig. S1). $\beta$-Actin is mainly located in protrusions at the front of polarized T-cells. Only small amounts of $\beta$-actin are detectable in the uropod (Fig. S1A). Indeed we observed only in $22 \pm 5\%$ of the resting cells ($n = 3$) and in $33 \pm 5\%$ ($n = 3$) of the stimulated cells weakly positive PLA signals ($1 \pm 0$ dots/cell; analysed in 68 cells with positive PLA derived from 3 independent experiments) for $\beta$-actin and flotillin-2. Both the % of cells with positive PLA and the number of dots per cell are thus clearly lower for this pair of antibodies as compared to the data shown in Figs. 1–5. In stimulated cells these dots were located outside of the uropod in 67 out of 68 inspected cells with positive PLA, derived from 3 independent experiments (Fig. S1B). Considering the uropod, this is thus a control showing zero response. The few dots detected outside of the uropod may be explained by an incidental close contact of occasional flotillin molecules located at the plasma membrane outside of the uropod with the abundant $\beta$-actin. The negative control with only the anti-$\beta$-actin antibody is shown in Fig. S1C.

### Interactions of PIPKI$\gamma$90 with P-ERM studied using PLA

GFP-tagged PIPKI$\gamma$90 accumulates in the uropod of murine T-cells (*Lokuta et al., 2007*; *Mathis et al., 2013*). We expressed an N-terminally HA-tagged PIPKI$\gamma$90 construct in human T-cells and observed colocalization of this construct with P-ERM in the uropod (Fig. 5A). We studied possible interactions of PIPKI$\gamma$90 with P-ERM using the PLA. As we have no highly specific antibodies to PIPKI$\gamma$90 available which work well in immunofluorescence staining, we transfected cells with PIPKI$\gamma$90 tagged with HA. The PLA was carried out with murine anti-HA antibodies and rabbit antibodies reacting with P-ERM. To visualize the transfected cells, we incubated the fixed cells with a FITC-tagged anti-murine IgG antibody that detects the HA antibody, together with the PLA probes. The results are shown in Fig. 5B (top panels: lower magnification; bottom panels: high magnification). We observe an extensive close association of PIPKI$\gamma$90 with P-ERM at the plasma membrane in $98 \pm 1\%$ of the transfected resting cells (range: 9–24 dots per cell; mean: $15 \pm 1$ dots per cell derived from 23 cells in 4 experiments). Similarly the PLA was positive in $89 \pm 7\%$ ($n = 3$) of the transfected polarized cells, mostly restricted to the uropods and lining the uropods in $75 \pm 7\%$ ($n = 3$) of the cells (Fig. 5B). The remainder of the polarized PLA-positive cells featured 1–2 dots/uropod. Dots indicating a positive PLA occurred exclusively in the transfected cells as can be appreciated from the top panels, where both transfected and untransfected cells are shown in the same picture (Fig. 5B). Negative controls carried out only with the anti-HA antibody are shown in Fig. 5C. A less extensive interaction was detected for PIPKI$\gamma$90 and flotillin-2 (data not shown).

## DISCUSSION

In polarizing T-cells, specific signaling molecules, adhesion receptors, raft-associated proteins and cytoskeletal proteins segregate into the uropod, a site where protrusion formation is locally suppressed, possibly by a membrane cytoskeletal scaffold (*Sánchez-Madrid & Serrador, 2009*). The exact mechanism of uropod scaffolding is as yet unknown. We have recently shown that flotillins as well as activated phosphorylated ERM proteins cooperate in

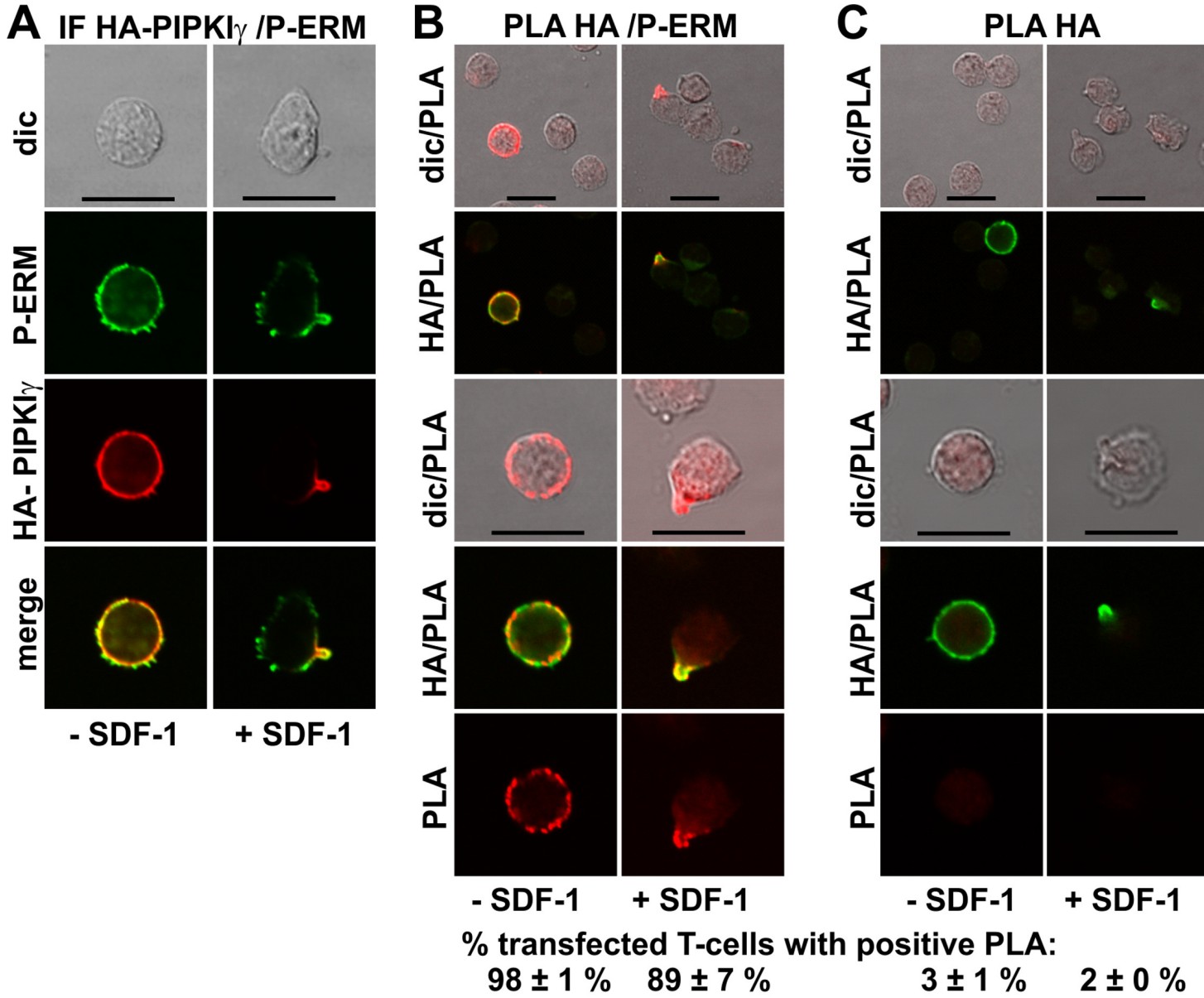

**Figure 5 Interaction of PIPKIγ90 and P-ERM in human T-cells studied with PLA.** (A–C) T-cells were transfected with HA-PIPKIγ90. Four h later the T-cells were preincubated for 30 min at 37°C, followed by a further incubation for 15 min without or with 40 ng/ml SDF-1, fixation with TCA and staining for HA (monoclonal murine antibody) and P-ERM (polyclonal rabbit antibody), followed by incubation with (A) fluorescently labeled second anti-murine (red fluorescence) and anti-rabbit IgG antibodies (green fluorescence) or (B) FITC-conjugated anti-mouse IgG, in order to visualize the transfected cells (green fluorescence), and PLA probes minus and plus, ligation and amplification (red fluorescence). (C) For negative controls, transfected cells were treated as described for (B) except that the anti-P-ERM antibody was omitted. The pictures are representative of 3 experiments. For (B) and (C), the top panels are overviews at lower magnification whereas in the lower panels single cells are shown at higher magnification. The percentage of transfected cells with one or more red fluorescent dots per cell was determined for 100 cells per sample and experiment (mean ± sem for 3 experiments). Scale bars, 10 μm.

T-cell uropod formation and that they mutually enhance their uropod capping (*Martinelli et al., 2013*). This could be explained by direct or indirect interactions of these proteins. We have now used the PLA approach to study close associations of uropod-located proteins. This assay is very sensitive and specific and provides a high signal to noise ratio. In contrast to FRET it can be used to study close proximity of endogenous unmodified proteins in situ. Positive results obtained with the PLA indicate either very close proximity of proteins for example in the same microdomains or direct interactions. The question arises whether the intensity of the PLA signal is proportional to the extent of protein association. This has been addressed in a study analyzing ErbB2 homoassociation in cell lines comparing FRET and PLA (*Mocanu et al., 2011*). The authors used flow cytometry for quantification of PLA, correlated to the extent of labeling with fluorophore-conjugated antibodies. They observed a non-linear dependence of the PLA signal on the extent of protein expression and association, whereas the FRET data showed a linear correlation. Steric hindrance between densely packed proximity probes may prevent enzymes from taking part in the amplification process. The authors recommend the PLA approach as a semiquantitative tool for measuring in situ protein associations (*Mocanu et al., 2011*). We thus did not attempt quantification of the PLA data by flow cytometry. Evaluation of the number of dots per cell is made difficult by the fact that coalescence of dots occurs, especially in the uropod but also in resting cells. The number of dots detected obviously will also depend on the expression levels of the interaction partners. Specificity of the reaction is indicated by the loss of signal when only one of the antibodies is used for PLA. Moreover we could detect no PLA signals in the uropod when we used antibodies to flotillin-2 and $\beta$-actin, the latter protein being located mainly outside of the uropod (Fig. S1).

Using FRET of differently tagged flotillin-1 and -2 expressed in human T-cells we previously showed that these proteins form heterodimers in both resting and chemokine-stimulated cells, and we can now confirm these findings for the endogenous flotillins using PLA (Fig. 1B). Another positive control is the positive PLA observed for P-ERM and PSGL-1 in resting and activated T-cells (Fig. 2B), in agreement with the literature indicating direct interaction of these proteins obtained by in vitro methods (*Ivetic & Ridley, 2004*).

We now provide novel data using in situ PLA in fixed human T-cells on close proximity of specific uropod components. We show that flotillins appear to closely associate with PSGL-1 in resting and stimulated T-cells, the reaction being restricted to the border of the uropod in polarized cells, confirming and extending the data obtained previously in human neutrophils using co-immunoprecipitation (*Rossy et al., 2009*). Flotillins and P-ERM appear to interact less extensively. Flotillins could thus indirectly recruit P-ERM to the uropod via interactions with PSGL-1.

Another T-cell uropod component is the enzyme PIPKIγ90, which synthesizes phosphatidylinositol-4,5-bisphosphate (PIP-2) and is involved in regulating uropod retraction in leukocytes (*Lokuta et al., 2007*; *Mathis et al., 2013*). PIP-2 is required for ERM activation, inducing a conformational change and allowing subsequent C-terminal phosphorylation (*Niggli & Rossy, 2008*). Uropod-localized PIPKIγ90 could thus result in locally enhanced PIP-2 production and ERM activation. Here we provide novel data

suggesting that PIPKIγ90 and P-ERM are in close proximity in resting and activated T-cells. Interestingly, the isoform PIPKIβ has also been reported to be targeted to the uropod of polarized neutrophils and to interact in vitro via an 83-amino acid C-terminal domain with EBP50 (ERM-binding phosphoprotein 50). These 83 C-terminal amino acids are not homologous in PIPKIγ90 isoforms (*Lacalle et al., 2007*). PIPKIβ and PIPKIγ90 may thus directly or indirectly interact with ERM proteins via differently structured binding sites.

In summary our data suggest either direct interactions of flotillins with PSGL-1, and of PIPKIγ90 with activated ERM proteins in resting and chemokine-activated T-cells and/or the presence of these uropod-located proteins in the same membrane microdomains. Preformed complexes of these proteins present already in resting cells could thus coalesce upon cell stimulation into the uropod. Whether P-ERM and PIPKIγ90 or flotillins and PSGL-1 indeed interact directly will have to be verified with other techniques. If so, the binding domains involved in interactions of flotillins with PSGL-1 and of PIPKIγ90 with P-ERM, and the possible regulation of these interactions by posttranslational modifications will have to be explored.

### Abbreviations

| | |
|---|---|
| **ERM** | ezrin/radixin/moesin |
| **FRET** | fluorescence resonance energy transfer |
| **HA** | hemagglutinin |
| **PFA** | paraformaldehyde |
| **PIP-2** | phosphatidylinositol-4,5-bisphosphate |
| **PIPKIγ90** | phosphatidylinositol-4-phosphate 5-kinase type I γ90 |
| **PSGL-1** | P-selectin glycoprotein-1 |
| **PLA** | proximity ligation assay |
| **SDF-1** | stromal cell derived factor 1 |
| **TCA** | trichloroacetic acid |

## ACKNOWLEDGEMENTS

We thank Dr. Erwin Sigel for the critical reading of the manuscript, Dr. Anna Huttenlocher for the generous gift of the PIP5KIγ661 construct and Dr. C Chaponnier for the kind gift of the anti-β-actin antibody.

### Funding

This work was supported by the Swiss National Science Foundation (Grant No 3100A_129655/1, to Verena Niggli). The funder had no role in study design, data collection and analysis, decision to publish, or preparation of the manuscript.

## Grant Disclosures

The following grant information was disclosed by the authors:
Swiss National Science Foundation: Grant No 3100A_129655/1.

## Competing Interests

The authors declare there are no competing interests.

## Author Contributions

- Tommy Baumann conceived and designed the experiments, performed the experiments, analyzed the data, wrote the paper.
- Sarah Affentranger performed the experiments, analyzed the data.
- Verena Niggli conceived and designed the experiments, analyzed the data, wrote the paper.

## Supplemental Information

Supplemental information for this article can be found online at http://dx.doi.org/10.7717/peerj.186.

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
