# Peer review of "Analysis of close associations of uropod-associated proteins in human T-cells using the proximity ligation assay"

_PeerJ, doi:10.7717/peerj.186_

## Round 0.1 · original submission · Minor Revisions

I agree with the reviewers that this submission is largely clearly written and the experiments justify the conclusions, and that it represents an advance of current knowledge. However, there is a consensus view that a few minor experimental and manuscript points would need to be addressed prior to acceptance for publication, which fall under two main categories.

1. Experimental
Though the data look convincing, I agree with reviewer 2 that the inclusion of another negative control comprising the use of two primary antibodies to proteins that do not interact would increase confidence in the data.
2. Manuscript
I also agree with reviewer 1 that, although the manuscript is largely well-written, it could do with a proof-read to weed out minor mistakes. Examples here include the one highlighted by reviewer 2, I also noticed:
Line 42: remove 'anymore'.
Line 48: replace 'allows detection also of' with 'also allows detection of'.
Line 115: replace 'We now investigated' with 'In the present study, we have investigated' or 'Here, we have investigated'.
I also think some detailed clarification of the image analysis would also be helpful, especially given this is the focus of the paper. For example, in the materials and methods, it says 100 cells per condition were evaluated, but in the text some of the numbers corresponding to total cell number analysed don't tally with this figure (e.g. a total of 261 cells analysed for n=3 in Fig. 4B). Similarly, I think you should indicate if the data are derived from a confocal stack or from a representative optical section (I'm assuming it's the latter, but I'm not sure).

Reviewer 1 ·

Basic reporting

No comments.

Experimental design

No comments.

Validity of the findings

No comments.

Additional comments

Baumann et al. examine pair-wise proximity of proteins, such as flotillin-1 and 2, phosphorylated ERM, PSGL-1 and PIPKIγ90, that are important for structuring the uropod during T-cell polarization. Previously, FRET and biochemical approaches have suggested associations among these proteins. However, FRET is based on over-expression while biochemical assays lack spatial information and are prone to post-lysis artifacts. The authors use the Proximity Ligation Assay to investigate associations of the endogenous proteins before and after T-cell stimulation. They find that flotillin-1/flotillin-2, flotillin-2/p-ERM, flotillin-2/PSGL-1 and PIPKIγ90/p-ERM complexes pre-exist in un-stimulated cells and become confined in the uropod upon stimulation. The results nicely confirm and extend previous findings.

The experiments were performed and interpreted well and presented clearly. While studies on the dynamics of these complexes will be needed to reveal the connections between the preformed complexes and the ones confined in the uropod , as it stands the work presented is an important first step. In addition, whether these associations are part of the same complex or separately co-exist represent an important future direction. For the time being, the authors may simply choose to further discuss these points.

Reviewer 2 ·

Basic reporting

The manuscript is generally well presented and easy to follow. I would recommend careful proof-reading since one or two mistakes are still present eg line 152 Fig1C should be 2C.

Experimental design

The use of the proximity ligation assay (PLA) adds information on the co-localisations seen between flotilins, PSGL-1, ERM and PIPKIgamma90 in the uropod of T-cells. The data presented is convincing as far as it goes; the minimum controls are shown for the the PLA assays (leaving out one of the two antibodies) but it's still difficult to really assess what's required for the PLA assay to give a positive signal eg only beta-actin and flotillin2 are shown as an 'irrelevent pair' and they do show significant interaction.

Validity of the findings

The data are robust with the required number of repetitions and statistical analysis.

Additional comments

Overall, I think this manuscript makes a modest advance over previous work published by this group and others but does not move us on much conceptually in terms of how close associations between proteins in the uropod are created.

---

## Round 0.2 · accepted · Accept

Dear Prof. Niggli,

Many thanks for your response and clarifications. Many congratulations on your interesting and informative paper: I enjoyed reading it. Have an excellent day!